# Identification and Characterization of PIWI-Interacting RNAs in Spinyhead Croakers (*Collichthys lucidus*) by Small RNA Sequencing

Qun Ji [1,2,†], Zhengli Xie [3,†], Wu Gan [1,2], Lumin Wang [1] and Wei Song [1,2,*]

1 East China Sea Fisheries Research Institute, Chinese Academy of Fishery Sciences, Shanghai 200090, China
2 College of Fisheries and Life Sciences, Shanghai Ocean University, Shanghai 201306, China
3 Fishery Machinery and Instrument Research Institute, Chinese Academy of Fishery Sciences, Shanghai 200090, China
* Correspondence: songw@ecsf.ac.cn
† These authors contributed equally to this work.

**Abstract:** PIWI-interacting RNAs (piRNAs) are an emerging class of small RNAs which protect the animal germline genome against deleterious transposable elements. Nevertheless, the characteristics and sex-related expression patterns of piRNA in *Collichthys lucidus* remain unknown. In this study, we first performed systematic next-generation high-throughput sequencing in *C. lucidus* ovaries and testes. We identified 3,027,834 piRNAs across six gonad libraries. Of these, 2225 piRNAs were differently expressed between testes and ovaries; 1195 were upregulated and 1030 downregulated in the testes. Interestingly, the potential target genes of 208 differentially expressed piRNAs had sex-related functions, including germ cell development, gonad development, ovarian follicle development, gamete generation, spermatid development, and spermatogenesis. Moreover, these target genes are involved in the TGF-β, Wnt, MAPK, mTOR, VEGF, and PI3K-Akt pathways. Further, 10 piRNAs were derived from *Nectin2* and *Mea1*, which play important roles in sexual reproduction, male gamete generation, and germ cell development. We also identified 5482 piRNA clusters across the gonads, among which 139 piRNA clusters were uniquely expressed in the testes and 98 in the ovaries. The expression of core sex-related piRNA was validated by real-time PCR. Overall, our findings provide significant insights into *C. lucidus'* sex-related piRNAs.

**Keywords:** piRNAs; piRNA clusters; fish; high-throughput sequencing; sex-related mechanism





## 1. Introduction

Spinyhead croakers (*Collichthys lucidus*), belonging to the order Perciformes, mainly inhabit the shore waters of the northwestern Pacific Ocean [1]. Owing to its excellent meat quality, *C. lucidus* is highly appreciated by consumers, and has become an economically important fish species [2]. Currently, its population size is decreasing due to overfishing, and the produce of this fish has failed to meet the market demand [3]. Therefore, it is necessary to explore the molecular mechanisms underlying reproduction and sex determination to stably maintain the population. The recently reported chromosome-level reference genome of *C. lucidus* allows investigation of the population genetics and performance of sex-determination studies [4]. De novo transcriptome sequencing showed 1288 genes differentially expressed between ovaries and testes; the differentially expressed genes were primarily involved in sexual reproduction and sex differentiation [5]. Additionally, the sex-related gene *vasa* had been shown to be conserved in *C. lucidus* and may be associated with germ cell development [6]. Nevertheless, the sex-related molecular mechanism in *C. lucidus* remains unclear.

PIWI-interacting RNAs (piRNAs) are germ cell-restricted small RNAs ranging from 25 to 32 nucleotides (nt) in length [7]. piRNA biogenesis involves primary and secondary

biogenesis, also known as the "ping-pong" pathway cycle [8,9]. piRNA regulatory mechanisms are well known for their role in epigenetic silencing of transposons and other repetitive elements [7]. A recent study proposed that piRNAs regulate mRNA and long non-coding RNA (lncRNA) expression by silencing transposons and post-transcriptional regulation in the germline [8,10]. Furthermore, piRNAs guide the cleavage of complementary mRNA predominantly at position 10 from the 5′ end of piRNAs [11]. A fundamental component of spermatogenesis, piRNAs ensure male fertility and genome integrity in multiple animals [10,12]. A recent study revealed that a lack of piRNAs results in progressive fertility loss in Caenorhabditis via epigenetic silencing of histone genes [13]. Moreover, piRNAs interact with PIWI-proteins to form RNA–protein complexes and allow gonadal development and gametogenesis [14]. To date, piRNAs have been characterized in several teleosts, including Nile tilapia [15], Zebrafish [16], and *Paralichthys olivaceus* [17]. However, the expression patterns and characteristics of the piRNAs in *C. lucidus* remain unknown.

In this study, we first performed next-generation high-throughput sequencing to identify gonad piRNAs and piRNA clusters in *C. lucidus*, fully investigating the sequence features, distribution characteristics, and expression patterns of piRNAs in the ovaries and testes. Moreover, the core sex-related piRNAs were described by analyzing the functions of their target genes.

## 2. Materials and Methods

### 2.1. Sample Collection

Six adult *C. lucidus* (three females and three males) were obtained from the East China Sea in Ningde, Fujian Province, China. The male fishes had a mean body length of $12.07 \pm 1.25$ cm and body weight of $33.80 \pm 10.75$ g; the female ones had a mean body length of $12.05 \pm 1.50$ cm and body weight of $36.75 \pm 13.04$ g. Fish were euthanized via intraperitoneal injection of an excessive pentobarbital dose (150 mg/kg) and placed on ice for ovaries' and testes' dissection. Samples were immediately frozen in liquid nitrogen and stored under $-80\,^{\circ}$C until further use. Fish handling and sample collection were approved according to the relevant guidelines and regulations of the Committee for Laboratory Animal Research at Shanghai Ocean University (SHOU-DW-2020-056). All efforts were made to minimize fish suffering.

### 2.2. Small RNA Library Preparation and Sequencing

Total RNA was isolated from the ovaries and testes of *C. lucidus* with a Total RNA Isolation Kit (Foregene, Chengdu, China). RNA concentration was detected by Nanodrop 2000 (Invitrogen, Carlsbad, CA, USA), and RNA integrity was evaluated using Agilent 2100 Bioanalyzer (Agilent Technologies, Palo Alto, CA, USA). The purified RNA was used for library construction using the NEB Next Multiplex Small RNA Library Prep Kit (New England Biolabs, Carlsbad, CA, USA) following the manufacturer's protocol. Briefly, 3′ and 5′ adaptors were added to both ends of the RNA, and the first cDNA strand was synthesized by reverse transcription and amplified by PCR. Small RNA fragments were obtained by 8% SDS-PAG gel for sorting 15–50 nt fragments. The libraries were assessed by Agilent 2100 (Agilent Technologies, Palo Alto, CA, USA) and quantified by Qubit (Invitrogen, Carlsbad, CA, USA). A Hiseq2500 sequencing platform (Illumina, SanDiego, CA, USA) was used for single-ended sequencing.

### 2.3. piRNA Identification

The original sequencing data was uploaded to the NCBI SRA database, under data accession number PRJNA660860. To obtain clean sequence reads, the primer, adaptor sequences, and the low-quality base of the 3′ end were removed from the raw data with FastQC software. The clean sequencing data were annotated with the PFAM and MIBASE databases. After filtering rRNA, tRNA, small nuclear RNA (snRNA), small nucleolar RNA (snoRNA), and micro-RNA (miRNA) sequences, the remaining sequences were used for piRNA prediction. At present, the piRNABank database (http://pirnabank.ibab.ac.in,

accessed on 20 May 2020) only includes piRNA sequences for humans, mice, rats, flies, and Zebrafish, but not for *C. lucidus*. Therefore, the k-mer sequences of these five species were used as positive training set to predict *C. lucidus'* piRNAs as follows: the k-mer sequences from the piRNA sequences of different species (human, mouse, rat, fruit fly, and nematode) were used to construct a sequence structure-based motif classifier; the piRNA sequence feature classifier was established and used to predict piRNAs using machine learning algorithms.

During model construction, two training data sets were established: a positive set, the piRNA sequence set, and a negative set, the non-piRNA sequence set. The positive set contains piRNA sequences from five known species (rat, mouse, human, fruit fly, and nematode); the sequence data was downloaded from Noncode Version 2.0 database and NCBI (NEMATODE: GI222138841-222138290; Fruit Fly: GI157362817-157361675). The negative data set was collected from the Noncode Version 2.0 database, which collects numerous types of ncRNAs (including 861 species). Data are mainly derived from three ncRNAs: (1) ncRNAs obtained from literature mining; (2) confirmed data from GenBank, (3) experimentally verified data, including miRNA, piRNA, mlRNA, snoRNA, snRNA, transfer-messenger RNA (tmRNA), signal recognition particle RNA (SRP RNA), guide RNA (gRNA), stem-bulge RNA (sbRNA), and snRNA-like RNA (snlRNA). Finally, a total of 34,675 ncRNAs were used as the negative sequence set of piRNA.

### 2.4. Differential Expression Analysis, Target Gene Prediction, and Functional Analysis

To analyze differentially expressed piRNAs between testes and ovaries, the putative piRNA sequence from each sample was mapped to the piRNA library, and the transcripts per million (TPM) value was calculated with the formula: TPM = number of piRNAs mapped to total reads/total sample read $\times 10^3$. The difference was considered significant for $|\log 2FC| > 1$ and $q$-value $< 0.01$. The R package pheatmap was used to create a heatmap. Moreover, the target genes of differentially expressed piRNAs were predicted by miRanda algorithm. The orthologous of the piRNA targets were identified by Blastn search. Then, enriched Gene Ontology (GO) terms and Kyoto Encyclopedia of Genes and Genomes (KEGG) were determined using a hypergeometric test.

### 2.5. piRNA Cluster Identification

piRNAs usually form clusters on chromosomes, with a length of about 20–100 kb, at a density between 40 and 4000. The piRNA cluster is first transcribed into a long single chain precursor, and further processed to form a mature piRNA. We used Protrac 2.0.5 software to identify piRNA clusters.

### 2.6. Validation of Differentially Expressed piRNAs by Real-Time PCR

We verified the differentially expression value of six sex-related piRNAs using real-time PCR. In brief, total RNA was extracted from the six gonad samples as described above. We used 1 μg of total RNA as template for the first-strand cDNA synthesis using a High-capacity cDNA Reverse transcription kit (Thermo Scientific, Waltham, MA, USA). The primers of the six piRNAs and two *piwil* genes were designed by Primer Premier 6.0 Software (Premier Biosoft International, Carlsbad, CA, USA, Table S1). Real-time PCR was performed with 10 μM primers, using SYBR Green Master Mix kit (Roche, Leverkusen, Germany) for piRNA and genes and run in an ABI Q6 system (Applied Biosystems Inc., Foster City, CA, USA). The melting curve analysis for the exclusion of primers combinations forming primer/dimers and specificity confirmation of newly designed primers was performed, and the melting curve s of all primers were single peak. We used u6 and GAPDH as internal reference for piRNA and *piwil* genes, respectively. Each experiment was performed using three biological and three technical replicates. The data was calculated using the $2^{-\Delta\Delta CT}$ method [18]. Student's *t*-test was used to evaluate statistical differences between testes and ovaries, at a $p < 0.05$ threshold. The statistical analysis and graphical construction were performed using GraphPad Prism 8.0.2.

## 3. Results

### 3.1. Characteristics of Gonad piRNAs in C. lucidus

piRNA is a kind of small RNA, which plays vital roles in the regulation of mRNA expression in germline. For the piRNA cluster, is a cluster that formed by different piRNAs on the chromosomes, which is the source for the mature piRNA. Therefore, we focus on the piRNAs and piRNA clusters. To investigate sex-related piRNAs, we performed small RNA sequencing in the ovaries and testes of *C. lucidus*. We obtained 30,747,399–38,288,768 clean reads from each of the six samples; testis samples displayed more clean data (mean = 36,740,960 reads) compared with ovary samples (mean = 33,907,586 reads, Table S2). On average, 4,408,545 clean reads (12.00%) were mapped to piRNAs in each testis sample, and 5,695,834 (15.48%) in each ovary sample. piRNA sequences ranged from 25 to 32 nt in length, and the peat occurred at 27 nt (Figure 1A). Moreover, the base bias for most piRNA was consistent, showing a strong uridine bias at the 5'-end, and an obvious preference for adenine at the 10th position in both ovary and testis (Figure 1B,C). The piRNAs were unevenly distributed among chromosomes; the distribution density was not proportional to the chromosome length (Figure 1D).

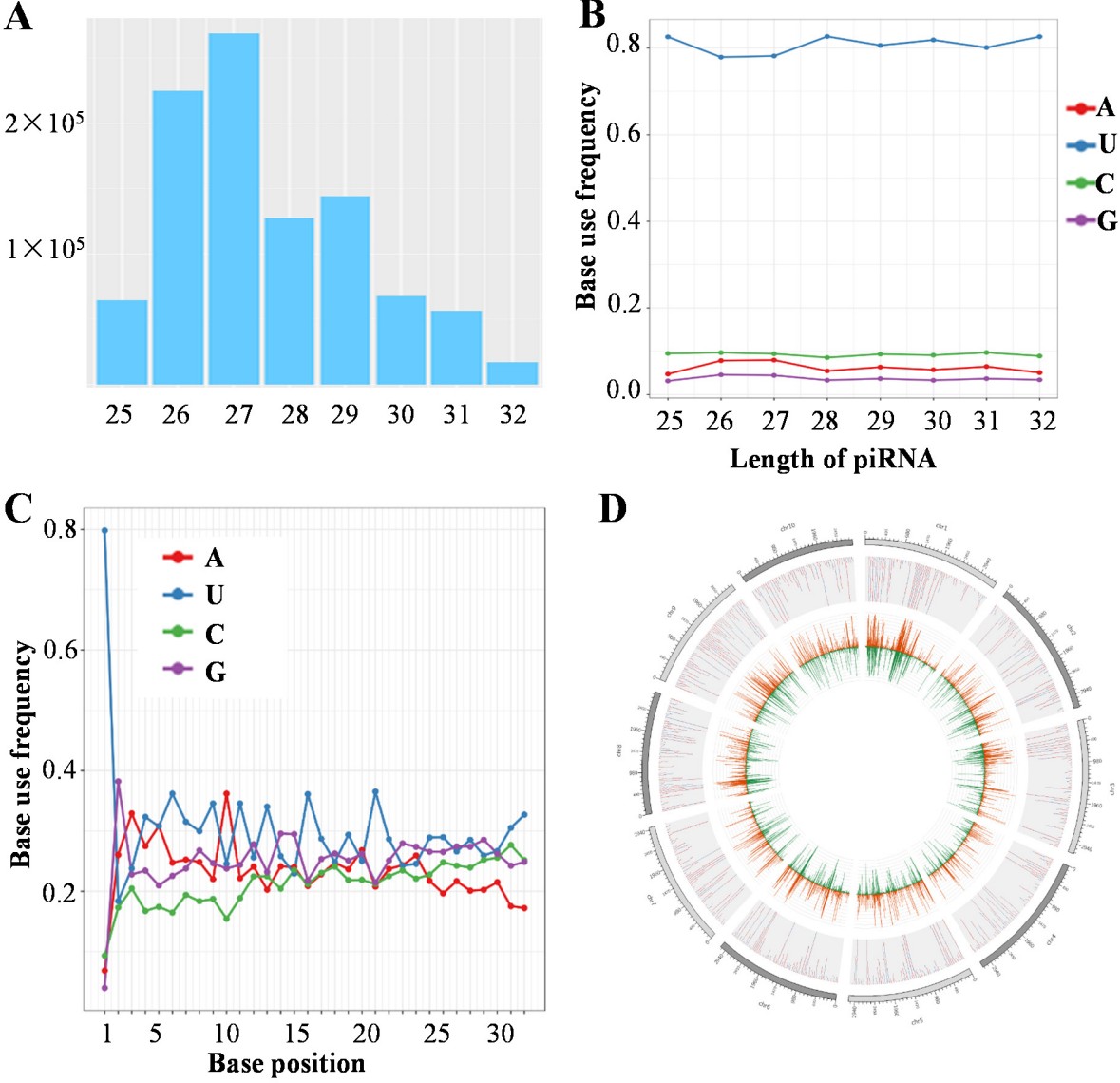

**Figure 1.** *Cont.*

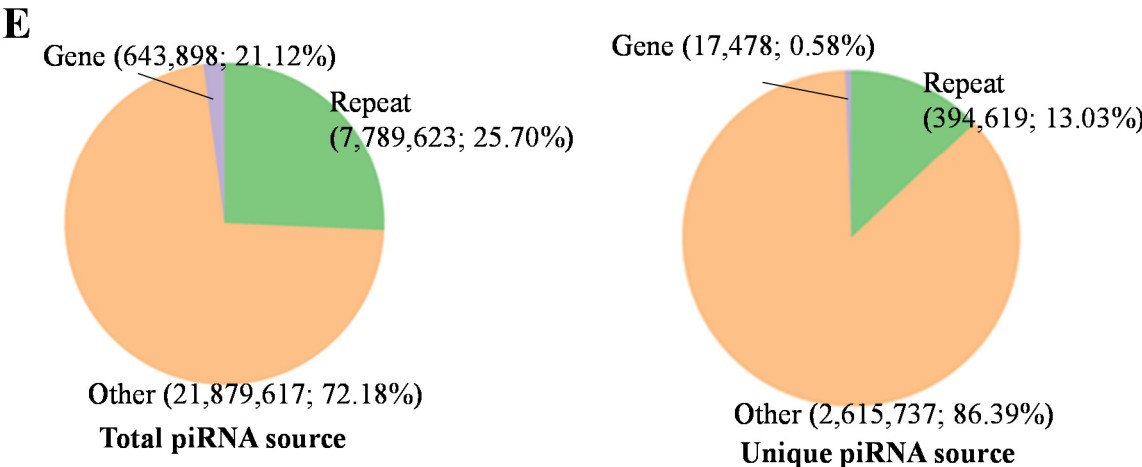

**Figure 1.** Characteristics of piRNAs in *C. lucidus* gonads. (**A**) Length distribution of piRNA sequences. (**B**) 5'-end base bias for different piRNA lengths. (**C**) Base bias at different base positions. (**D**) piRNA distribution in genomic regions (only showing the typical 10 chromosomes). The outer circles indicate chromosomal locations, and inside circles with red and green peaks indicate the piRNA distribution. (**E**) Venn diagram showing the piRNAs mapped to transposons and genes.

To characterize the piRNA distribution in genomic regions, the putative piRNAs were mapped to transposon and gene sequences (Figure 1E). For all piRNAs, the read number of piRNAs derived from repeat sequence regions, including transposons (7,789,623, 25.70%) was higher than that from gene regions (643,898, 2.12%). Similarly, the unique piRNA sequence genome distribution showed that 394,619 (13.03%) reads were aligned to repeat sequence regions, and only 17,478 (0.58%) were mapped to gene regions.

### 3.2. Differentially Expressed piRNAs

A total of 3,027,834 unique putative piRNAs were identified across all six libraries. Of these, 1,713,442 putative piRNAs were uniquely expressed in testis samples, 858,029 were only found in ovary samples, and 456,363 were shared between ovary and testis (Figure 2A). Among the latter, significantly differentially expressed piRNAs were identified using a strict filtering parameter of $q$-value < 0.01 and $|\log2 \text{(fold change)}| > 1$. We found 2225 piRNAs differently expressed between testis and ovary samples; 1195 were upregulated and 1030 were downregulated in the testes compared to the ovaries (Figure 2B).

### 3.3. Putative Functions and Pathways of Differentially Expressed piRNAs

To explore the functions and pathways of the differentially expressed piRNAs, we first predicted the target genes of these piRNAs using miRanda. A total of 8737 target genes were observed for the 2225 gonadal differentially expressed piRNAs, generating 248,810 potential piRNA-target gene pairs. For the top 50 piRNA-target pairs (Figure S1), three piRNAs (uniq-1562774, uniq-1511231 and uniq-1521889) targeted the same six genes, including Janus kinase 2 (JAK2), ETS transcription factor ELK3 (ELK3), GTP binding protein 2 (Gtpbp2), High choriolytic enzyme 1 (Hcea), MINDY lysine 48 deubiquitinase 1 (Mindy1), and ATPase Na$^+$/K$^+$ transporting subunit beta 2 (ATP1B2). GO analysis of all predicted target genes revealed their involvement in protein folding, carbohydrate metabolic process, proline biosynthetic process, nucleotide-excision repair, and tRNA modification (Figure 3A). Interestingly, 208 differentially expressed piRNAs might be involved in sex-related functions, including germ cell development, gonad development, ovarian follicle development, gamete generation, spermatid development, and spermatogenesis, by regulating deleted in azoospermia-like (Dazl), anti-Mullerian hormone (Amh), bone morphogenetic protein 15 (Bmp15), spindlin 1 (Spin1), piwi-like RNA-mediated gene silencing 1 (Piwil1), testis development related protein (Tdrp), and male-enhanced antigen 1 (Mea1)

genes (Figure 3B, Table S3). The KEGG analysis showed that the target genes were mainly associated with the glycine, serine, and threonine metabolism, oxidative phosphorylation, complement and coagulation cascades, spliceosome, and mRNA surveillance pathway (Figure S2A). Additionally, we found that multiple target genes were involved in sex-related pathways, including the TGF-β [19], Wnt [20], MAPK [21], mTOR [22], VEGF [23], and PI3K-Akt signaling pathways (Figure S2B) [23].

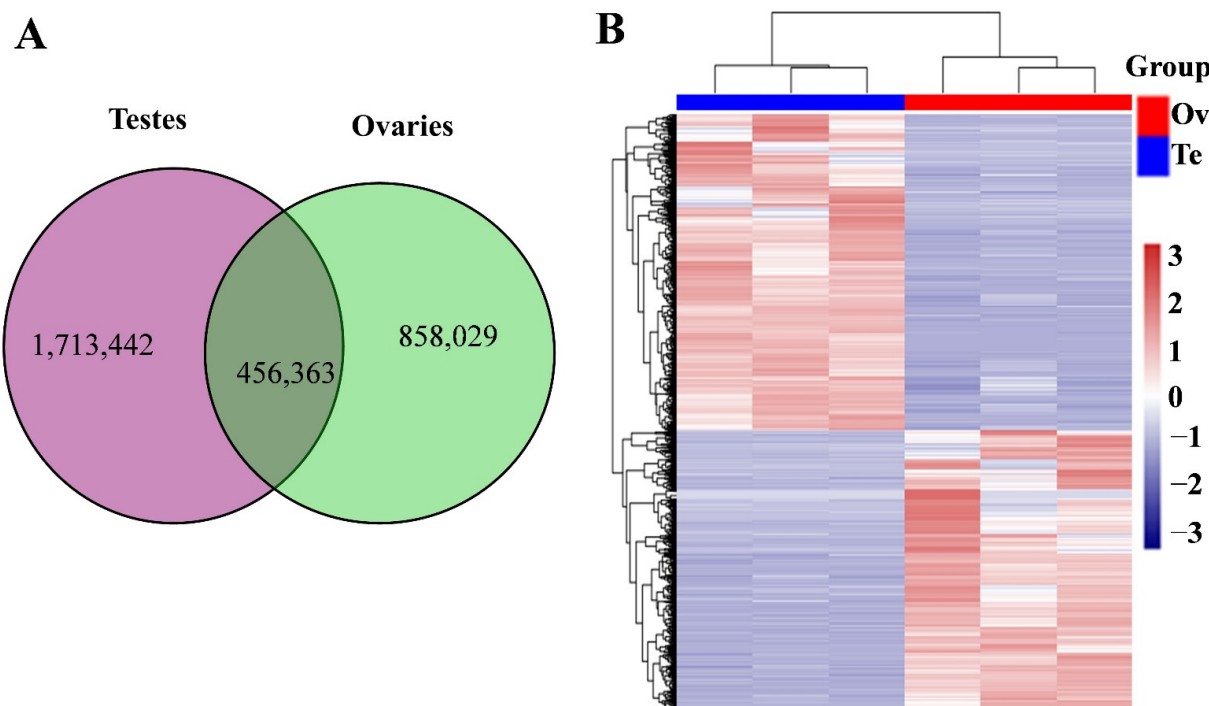

**Figure 2.** piRNA expression pattern in testes and ovaries. (**A**) Venn diagram showing shared and unique piRNAs in testes and ovaries. (**B**) Heatmap showing the significantly different piRNA expression between testes (Te) and ovaries (Ov). Columns indicate samples and rows indicate piRNAs. Red indicates high expression and blue indicates low expression.

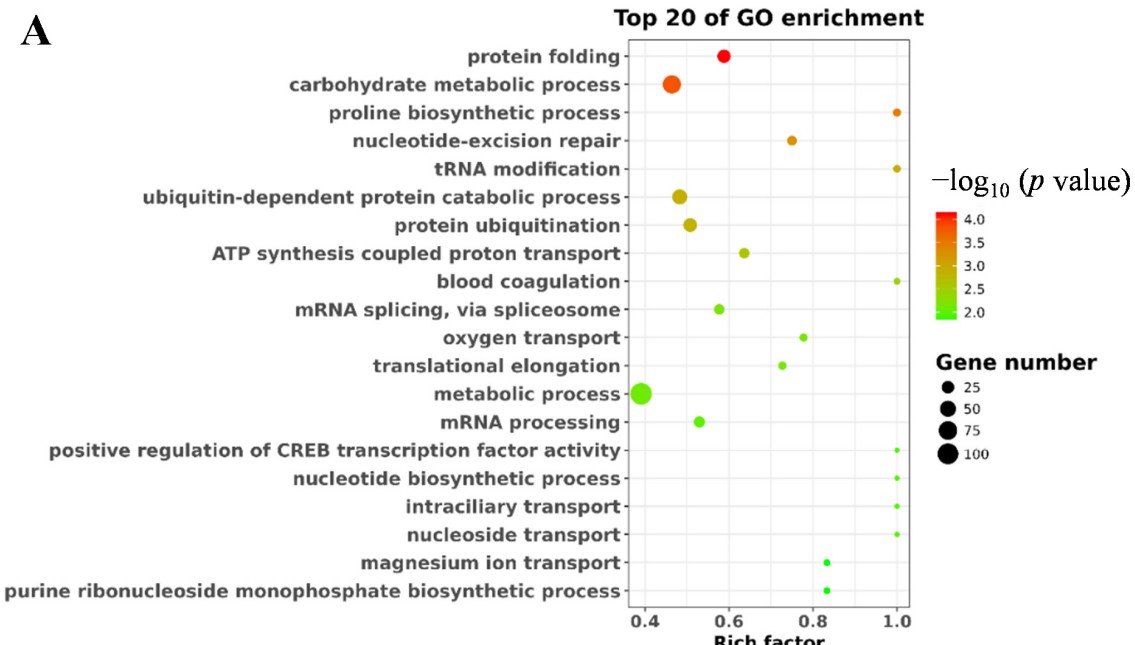

**Figure 3.** *Cont.*

**B**

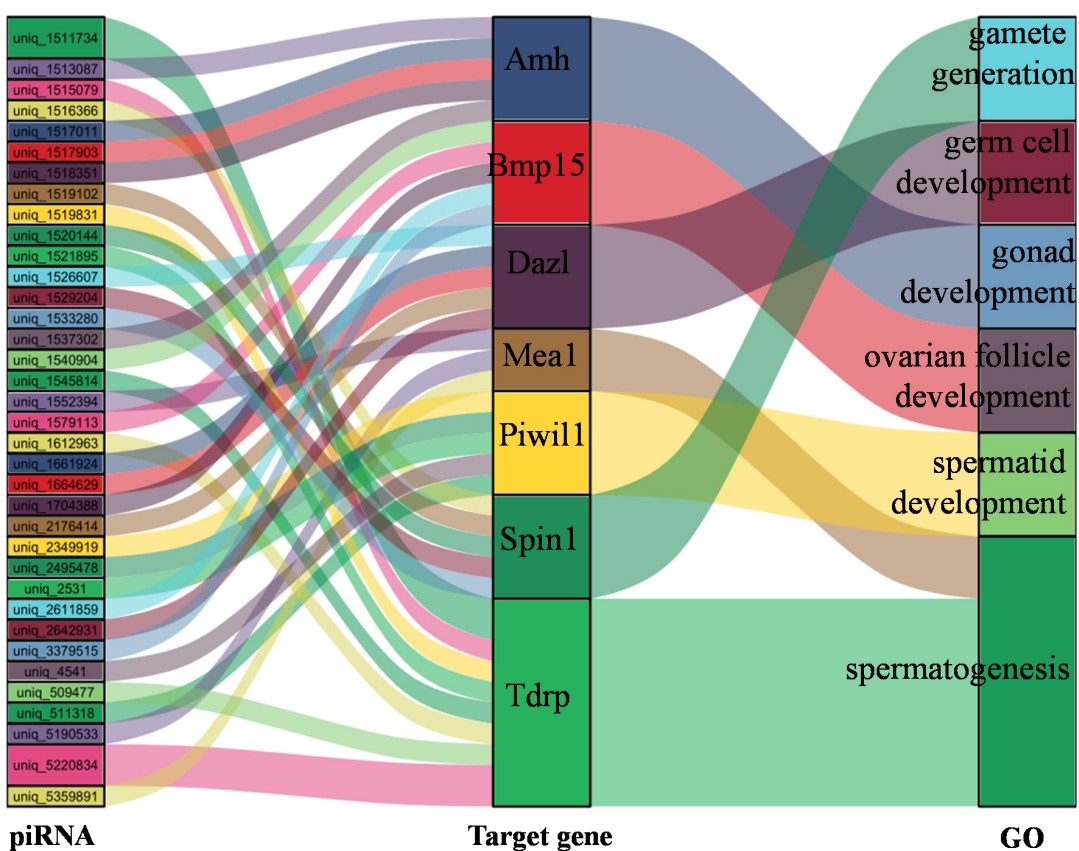

**Figure 3.** Putative functions and pathways of differentially expressed piRNAs. (**A**) GO analysis of the target genes of differentially expressed piRNAs. Only the top 20 GO terms are presented. Bubble color indicates the *p* value, and bubble size indicates the number of enriched genes. (**B**) Sankey diagram showing the relationship between differentially expressed piRNAs, target genes, and sex-related functions.

### 3.4. Candidate Host Genes of piRNAs

A total of 2329 host genes were identified for our 17,478 putative piRNAs. GO analysis of these host genes showed that within the category of biological process, host genes were mainly involved in protein phosphorylation, phosphorylation, cortical cytoskeleton organization, cortical actin cytoskeleton organization, and nucleoside triphosphate biosynthetic processes (Figure 4A, Table S4). To identify sex-related piRNAs, we searched for the associated GO terms using sperm, male, female, gender, sex, gonad, germ, and reproduction as keywords. Finally, we found 10 piRNAs derived from *nectin cell adhesion molecule 2* (*Nectin2*) and *Mea1* and involved sexual reproduction, male gamete generation, and germ cell development (Figure 4B).

### 3.5. Identification of piRNA Clusters

We identified 5482 piRNA clusters cross the gonads of *C. lucidus*, among which cluster 3735 (located at chr15:18653474-18901095) had the longest sequence length (Figure 5A, Table S5). Interestingly, 139 piRNA clusters were uniquely expressed in the testis and 98 piRNA clusters in the ovaries (Figure 5B, Table S6). The chromosome distribution of clusters revealed that piRNA clusters were non-uniformly distributed in different chromosomes, showing no proportionality to chromosome length (Figure 5C). Chromosome 24 displayed the highest distribution density of piRNA clusters, including 869 clusters. We next analyzed the piRNA data of each cluster, and found that piRNA cluster 678 included the largest piRNA number, with 9087 piRNAs (Figure 5D).

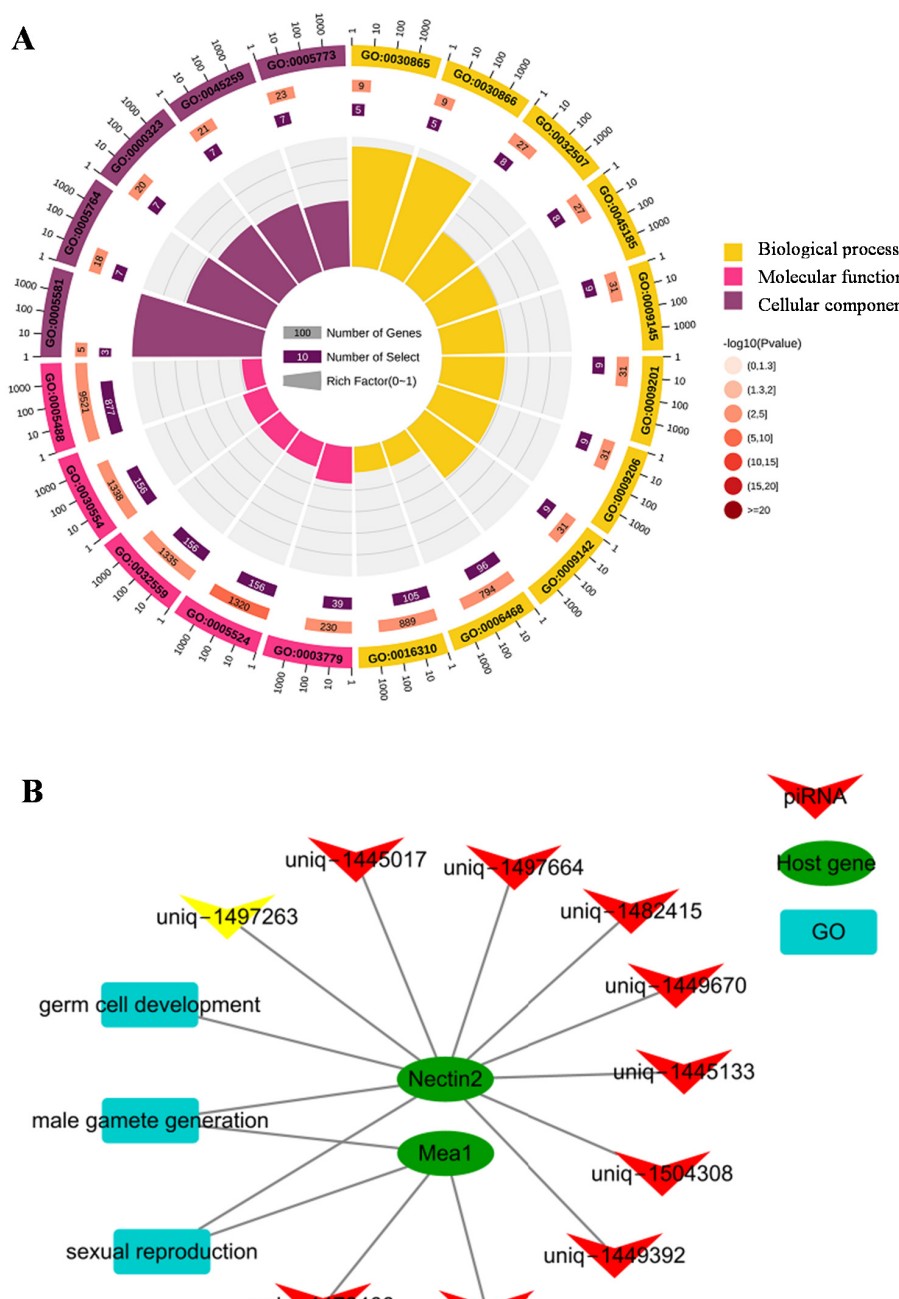

**Figure 4.** Candidate host-genes for piRNAs. (**A**) GO analysis of candidate host-genes of all putative piRNAs. From outside to inside: the first circle indicates the GO ID (yellow represents biological processes, pink molecular functions, and purple cellular components) and the scale shows the number of genes; the second circle shows the gene number of the GO terms; the third circle shows the number of target genes enriched in each GO term; and the fourth circle indicates the Richfactor value. (**B**) Network showing the relationship between piRNAs, host-genes, and sex-related functions.

### 3.6. Differential Expression Analysis of piRNA Clusters

For shared piRNA clusters in the ovaries and testes, differentially expressed piRNA clusters were identified using the strict filtering parameter of *q* value < 0.01 and |log2 (fold change)| > 1. We identified 1057 piRNA clusters differentially expressed between the ovaries and testes, including 545 upregulated and 512 downregulated in the testes compared with the ovaries (Figure 6A). A heatmap showed that the expression value of piRNA clusters was successfully divided into two clusters, namely testes and ovaries

(Figure 6B). To investigate the target genes of piRNA clusters, we identified genes located within 4 kb upstream and downstream of the piRNA clusters. A total of 3141 genes were found neighboring 2255 piRNA clusters, generating 3193 "piRNA cluster-gene" pairs. The top six piRNA clusters with the most target genes were cluster 1640, cluster 4850, cluster 678, cluster 3811, cluster 3515, and cluster 246, which affect 9–14 neighbor genes (Figure 6C).

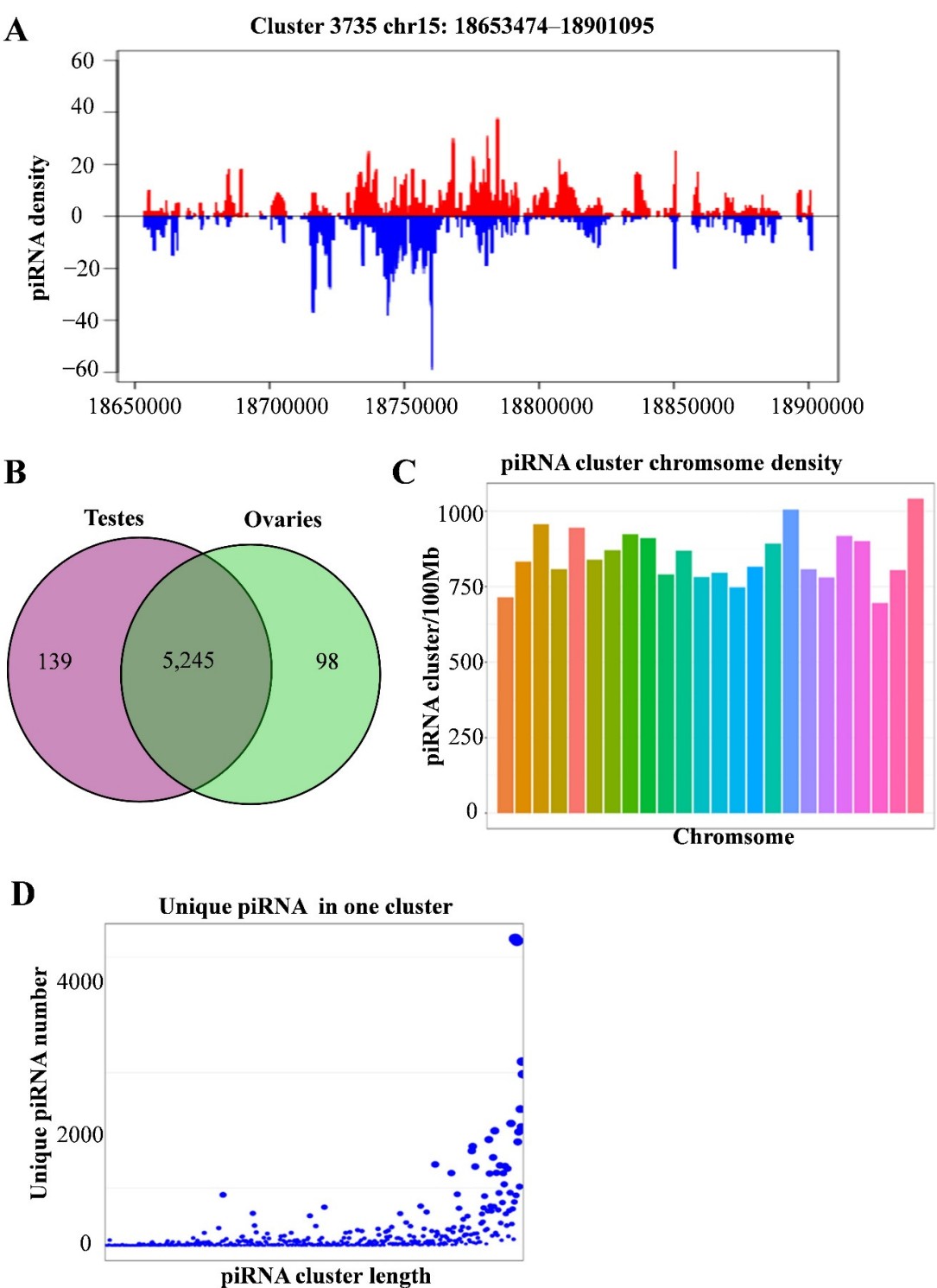

**Figure 5.** Characteristics of piRNA clusters. (**A**) piRNA density of cluster 3735, which has the largest clusters. (**B**) Venn diagram showing shared and unique piRNA clusters in the testes and ovaries. (**C**) Chromosome distribution of piRNA clusters. (**D**) Number of piRNAs in each piRNA cluster.

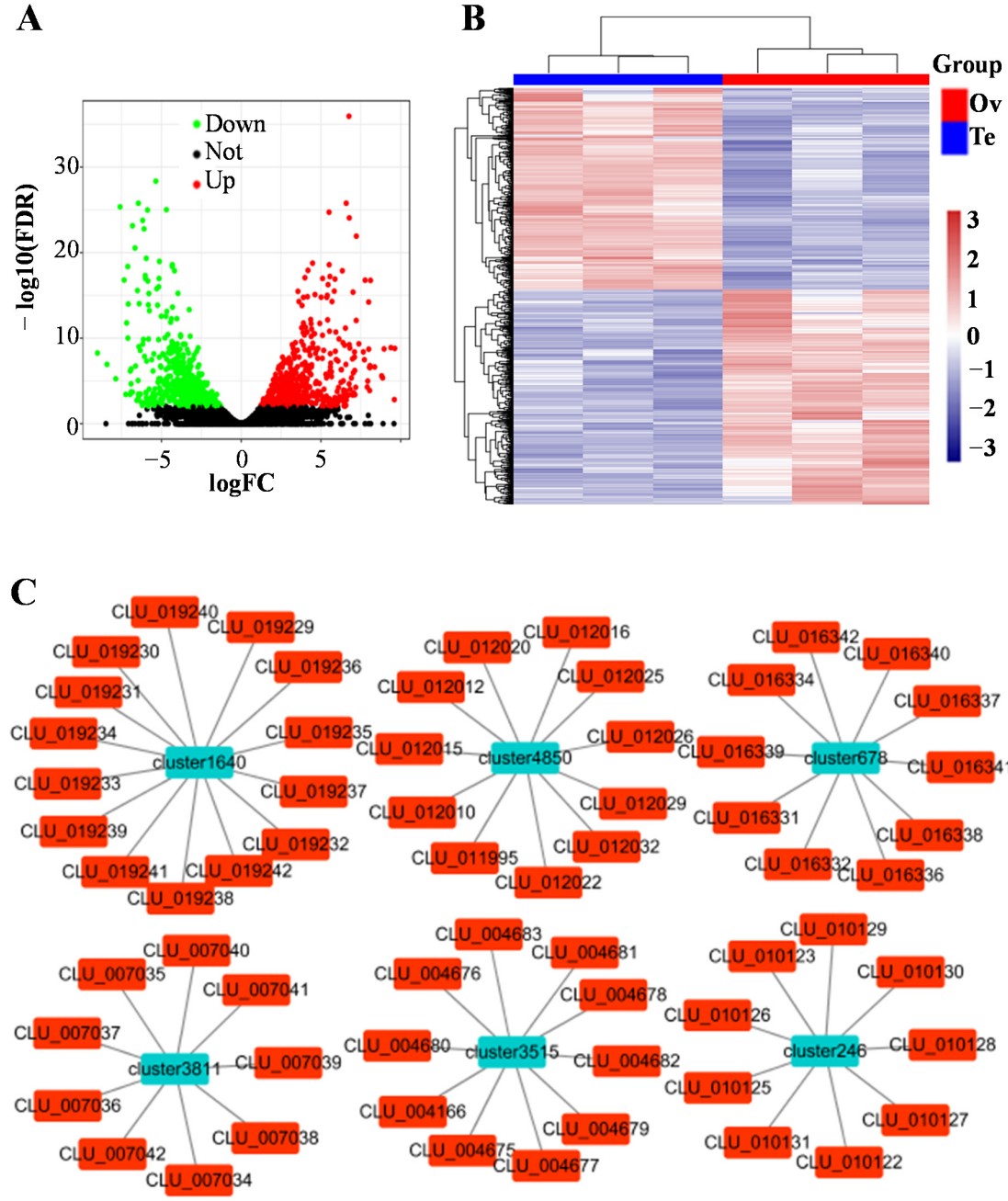

**Figure 6.** Differential expression analysis of piRNA clusters. (**A**) Volcanic plot displaying differentially expressed piRNA clusters between testes and ovaries. (**B**) Heatmap showing differentially expressed piRNA clusters between testes (Te) and ovaries (Ov). Columns indicate samples and rows indicate piRNAs. Red indicates high expression and blue indicates low expression. (C) Network between piRNA clusters and their nearby genes. Only the top six piRNAs with the largest number of adjacent genes are shown.

### 3.7. Validation of Differentially Expressed piRNAs

To identify the core piRNAs involved in sex regulation, we constructed a "piRNA—target gene—sex-related pathway" network (Figure S3). Based on the network, six piRNAs (uniq-1510639, uniq-1510671, uniq-1510715, uniq-1615793, uniq-1709642, and uniq-1615047) with high expression abundance and fold changes were selected for real-time PCR. Consistent with sequencing data, real-time PCR showed upregulated expression levels of uniq-1510639, uniq-1510671, and uniq-1510715 in the ovaries, while uniq-1615793, uniq-

1709642, and uniq-1615047 were upregulated in the testes (Figure 7A, Table S7). Except for uniq-1709642, the differences of all other piRNAs were significant. Moreover, two piwil proteins were also evaluated. Compared to other tissues (heart, liver, kidney, brain, muscle, eye, stomach, and intestines), Piwil1 and Piwil2 were both highly expressed in gonads; Piwil2 was primarily detected in the ovary (Figure 7B, Table S8).

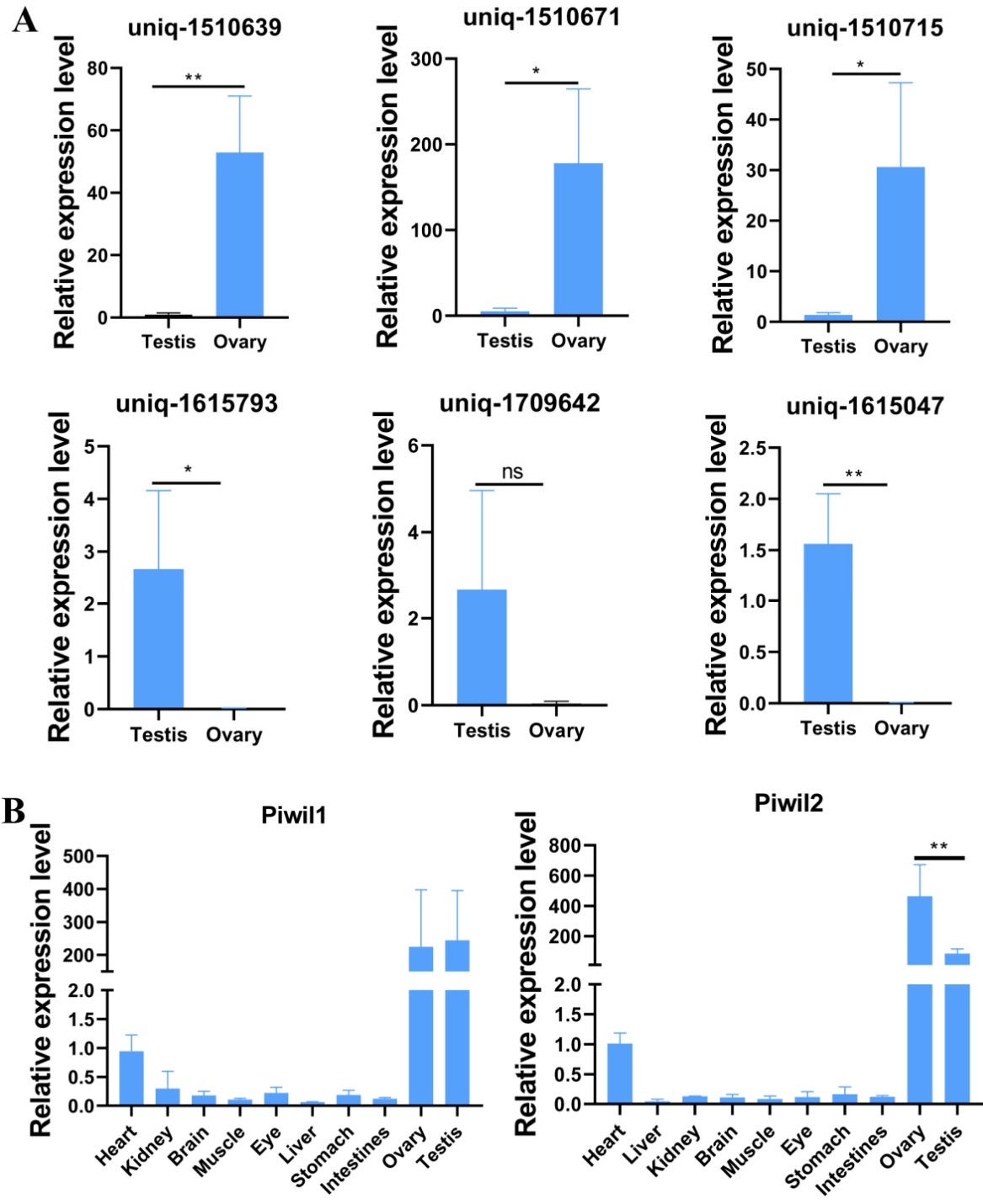

**Figure 7.** Validation of differentially expressed piRNAs. (**A**) Verification of six differentially expressed piRNAs by real-time PCR. $N = 3$, *t*-test. * $p < 0.05$, ** $p < 0.01$. (**B**) Expression of piwil1 and piwil2 detected by real-time PCR. N = 3. One-way ANOVA, ** $p < 0.01$ comparison between testes and ovaries, and $p < 0.01$ compared with testes and ovaries.

## 4. Discussion

Emerging evidence indicated that piRNAs play a fundamental role in various biological processes, including gametogenesis, sex determination, and early embryonic development [10,24–26]. piRNAs mechanisms involve regulating transposon silencing, genomic stability, mRNA decay, and translation inhibition [7,8]. Nevertheless, the expression pattern of gonad piRNAs in *C. lucidus* remains unknown. In this study, we first systematically identified gonad piRNAs and piRNA clusters in *C. lucidus,* showing the sequence features, distribution characteristics, and specific expression pattern of piRNAs in the ovaries and testis. Moreover, we illustrated their potential functions and pathways, identifying several key sex-related piRNAs.

As a novel class of small RNAs, piRNA is proverbially expressed in various animal cells, especially enriched in germ cells, and generally 24–32 nucleotides in length [8]. Consistent with the classical piRNA features, *C. lucidus* piRNAs were 25–32 nt long, with a peak at 27 nt. Additionally, piRNAs identified in human, rat, horse, Zebrafish, and other mammalian germ cells, show a strong base bias and are non-homogeneously distributed on different chromosomes [27–29]. Accordingly, the piRNAs obtained in *C. lucidus* showed a strong preference for uracil at the 5′ end, and for adenine at the 10th base. These sequence features have been associated with the ping-pong model of piRNA biogenesis [9]. In germ cells, piRNA biogenesis involves secondary amplification, achieved by Aub and AGO3 proteins-mediated ping-pong pathway, which is also a mechanism of post-transcriptional gene silencing [27]. The piRNAs had a 5′U bias at the 5′ end was loaded to Aub, and piRNAs have adenine at 10th position was bond to Ago3, and these piRNAs are antisense to transposable element mRNA and guide these two proteins to scavenge RNA targets [30]. AGO3 was previously identified in bony fish on chromosome ARUT59; it is similarly expressed in different tissues [31]. The typical 5′-end U and 10th adenine bias of the piRNAs in *C. lucidus* indicate their potential role in post-transcriptional regulation by guiding Aub and AGO3 proteins to degrade target mRNAs. Overall, these predicted piRNAs show consistent characteristics with previously described piRNAs, suggesting that we obtained true piRNAs.

In the present study, we applied a k-mer scheme to predict piRNAs. Compared with previous methods, this novel approach does not require a reference genome, having greater advantages for piRNA identification [32]. This way, we obtained 3,027,835 unique putative piRNAs from *C. lucidus* gonads. Compared to other aquatic animals, *C. lucidus* has more abundant piRNAs; only 4857 piRNAs were found in shark liver [33], 5865 in adult gonads of *Paralichthys olivaceus* [34], and 115,491 in mud crab *Scylla paramamosain* [35]. However, mammals usually transcribe more piRNAs; Mongolian horses express 4,936,717 piRNAs in the testes [27]. Moreover, more piRNAs were specifically expressed in the testes than in the ovaries in *C. lucidus*. piRNA expression in either testes or ovaries varies among species; several display more abundant piRNAs in the testes (such as Nile tilapia [15,36] and Zebrafish [16]) but others show the opposite effect (such as Chinese giant salamanders [37]). Therefore, this sex-bias in piRNA expression differs among species, which may lead to different piRNA functions and regulatory mechanisms among species. Furthermore, we identified 2225 differentially expressed piRNAs between *C. lucidus* ovaries and testes. Previous studies have shown that piRNAs have a profound impact on sex determination, differentiation, and development [25,26]. Their gonad-specific and differential expression patterns suggest that piRNAs might be involved in *C. lucidus* sex-related mechanisms. piRNA sequence information and expression data would be useful for investigating the sex determination, differentiation, and development in *C. lucidus.*

GO analysis revealed that 10 piRNAs derived from two genes (*Nectin2* and *Mea1*) were involved in sexual reproduction, male gamete generation, and germ cell development. *Nectin2* is a junction molecule, heterospecialized at the base and apex, contributing to the blood-testicular barrier and supporting sperm cell adhesion, and related to male infertility [38]. Knocking out the *Nectin2* or *Nectin3* gene leads to sterility in male mice; their sperm shows severe teratozoospermia, changed motility, and impaired ability to fertilize

eggs [39]. Similarly, *Nectin2* acts as an essential sexual maturation mediator and is involved in congenital hypogonadotropic hypogonadism [40]. *Mea1*, a male-enhanced antigen gene, which plays an important role in mammalian spermatogenesis and/or testicular development, is genetically conserved and specifically expressed at late stages of spermatogenesis in the testes [41,42]. Therefore, we speculate that these 10 piRNAs may play vital roles in spermatogenesis regulation in male *C. lucidus.*

KEGG analysis showed that the target genes of differentially expressed piRNA were involved in gametogenesis and reproduction-related pathways, including TGF-β [19], Wnt [20], MAPK [21], mTOR [22], VEGF [23], and PI3K-Akt signaling pathways [23]. The TGF-β pathway plays important roles in various biomedical programs, including reproductive, gametogenesis, and even sex determination. TGF-β is essential for the maintenance of oocyte meiotic arrest, enhancing natriuretic peptide type C levels in mouse granulosa cells [43]. Deactivation of TGF-β signaling via small molecules can effectively promote the proliferation of undifferentiated spermatogonia and recovery of spermatogenesis after chemotherapy [44]. *Amh*, a member of the TGF-β superfamily which inhibits female reproductive tract development, has been described in multiple vertebrates, particularly in fish, as an initiator regulator of gonad development and sex determination, [19,45]. Our findings provide the first evidence that gonad-specific differentially expressed piRNAs are involved in the TGF-β pathway, highlighting their potential role in sex-related mechanisms.

Wnt signaling is a canonical pathway necessary and sufficient to drive gonad development and spermatogenesis in various species [46]. Inhibition of Wnt signaling leads to a decrease in the number of germline stem cells and a delay in ovarian regeneration in Zebrafish [47]. In late ovarian, Wnt11 was mostly expressed in granulosa cells and oocytes, while Wnt9b was mainly expressed in granulosa cells; in testis, Wnt pathway-related genes were mainly detected during early spermatogenesis in rainbow trout [48]. Suppression of Wnt signaling enhances the maturation of Sertoli cells in idiopathic non-obstructive azoospermia patients, restoring the ability of these cells to support germ cell survival [49]. In Tambaqui (*Colossoma macropomum*), prior to differentiation, the components of the Wnt/β-catenin pathway, FOX and FST, determined female sexual development while antagonistic pathways marked male differentiation, suggesting the Wnt/β-catenin pathway is involved in sex differentiation, activating in females or antagonizing in males [50]. Our data showed that predicted target genes of piRNAs were involved in Wnt signaling; thus, these piRNAs might be crucial for gonad differentiation in *C. lucidus*. Further research is needed to clarify the functions and mechanisms of these piRNAs in *C. lucidus* sex-related processes.

## 5. Conclusions

Collectively, our findings first characterized the expression profile of piRNAs, and their potential target genes, function and signaling pathways in the gonads of *C. lucidus*. Our data indicated that piRNAs might be involved in various sex-related processes in *C. lucidus*. However, to better understand the role of piRNAs in sex-related processes, further research should focus on their regulatory mechanisms and functions in germ cells.

**Supplementary Materials:** The following supporting information can be downloaded at: https://www.mdpi.com/article/10.3390/fishes7050297/s1, Figure S1: Network of the top 50 piRNA-target pairs; Figure S2: Pathway analysis of target genes of differentially expressed piRNAs; Figure S3: A network established using piRNA, target genes, and sex-related pathways; Table S1: Primer sequence used in this study; Table S2: Overview of the sequencing data; Table S3: The relationship of differetially expressed piRNAs-target genes-sex related function; Table S4: GO analysis for the host genes of putative piRNAs; Table S5: The characteristics of piRNA cluster; Table S6: The piRNA clusters uniquely expressed in testes or ovaries; Table S7: Raw data of piRNA detected by real time PCR; Table S8: Raw data for piwil genes detected by real time PCR.

**Author Contributions:** W.S. and L.W.: Conceptualization, Funding acquisition, and original draft. Q.J. and Z.X.: Resources, Formal analysis, Data curation and original draft, Writing—review and editing. W.G.: Formal analysis and Visualization. All authors have read and agreed to the published version of the manuscript.

**Funding:** This work was supported by the Basic Research Fund for State-Level Nonprofit Research Institutes of ESCFRI, CAFS (2019M02); The earmarked fund for CARS-47; Central Public-interest Scientific Institution Basal Research Fund, CAFS (2020TD76). The key Project of Zhejiang Province of China (2020C02015); National Key R&D Program of China (2020YFD0900805).

**Institutional Review Board Statement:** The animal study was reviewed and approved by Academic Committee of Shanghai Ocean University (approval code SHOU-DW-2020-056, approved on 13 May 2019).

**Data Availability Statement:** The datasets used and/or analyzed during the current study are available from the corresponding author on reasonable request. The sequencing data has been uploaded into NCBI database, and the serial number was PRJNA660860.

**Conflicts of Interest:** The authors declare that they have no competing interests.

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
