# Peer review of "Identification and Characterization of PIWI-Interacting RNAs in Spinyhead Croakers (Collichthys lucidus) by Small RNA Sequencing"

_fishes, doi:10.3390/fishes7050297_

Round 1
Reviewer 1 Report
Manuscript fishes-1911334 is a comprehensive work providing identification and characterization of PIWI-interacting RNAs in Spinyhead croakers
Title
Ln 3 – Should be “ Spinyhead croakers (Collichthys lucidus)”
Materials and Methods
Lns 119-121 – A brief procedure for GO terms analysis and KEGG should be included!
Ln 131– State the concentration of primers used!
Lns 137 – a. “The data was calculated….” include the references in order to support this statement!
b. Did the authors conduct the melting curve analysis for the exclusion of primers
combinations forming primer/dimers and specificity confirmation of newly designed primers? State it in ms!
Ln 142 – a) Mention the GenBank accession number and the references where the primers designed from!
b) Target size for each pair of primers should be included in Table S1
Results
Ln 159 – Should be “ C. lucidus”
Lns 180-181– Mention the software used for creating a heatmap in M&M section!

Reviewer 2 Report
The proposed manuscript (ms) “Identification and characterization of PIWI-interacting RNAs in Collichthys lucidus by small RNA sequencing” greatly contributes to the study of molecular biology, especially function and signaling pathways of piRNA. I have not found any fundamental issues in the proposed ms. The ms is technically sound, presented in an intelligible fashion and written in good-quality English. Impact of the study is well highlighted. I appreciate the wide range of methods (RNA extraction, library preparation, NGS, data and statistical analysis, qPCR, gene expression). On the other hand, as the main discrepancy of the ms I consider that abbreviations are not clearly explained and used for better understanding.
After overall consideration of the manuscript quality I suggest major revision. After following the above recommendations, the manuscript can meet the requirements of the Fishes journal.
-
Abbreviations in the text are not explained after the first use, especially less known types of RNA. Maybee authors can use a list of abbreviations. Eg.: IncRNA,snRNA, snoRNA, miRNA, SRP RNA, ncRNA etc.
-
The study consists of two major analyses: RNA and RNA cluster analysis. Can authors better define differences between these two focuses and why they focused on them?
-
Row 40: What does the “vase” mean?
-
Row 69: Fishes or fish?
-
If you state the supplier’s place, be consistent. Sometimes, the state of the USA is included and sometimes it does not.
-
Row 98: Why did you select k-mer sequences from human, mouse, rat, fruit fly, and nematode? Please include this information into the ms.
-
Supplementary tables and figures are included in the main ms. In addition supplementary material has no legend/description.
-
Row 254: delete space in “valu e”
Round 2
Reviewer 2 Report
Authors followed all recommendations and the proposed manuscript was improved. I found another two points for improvements:
-
Table S1: modify primer sequences into the correct form and include 5’ and 3’ends.
-
Mention software in which all plots were generated.
After overall consideration of the manuscript quality I suggest minor revision. I am looking forward to seeing the published version of the paper.
